# The Counterattack of CNNs in Self-Supervised Learning: Larger Kernel Size might be All You Need

## Abstract

Vision Transformers have been rapidly uprising in computer vision thanks to their outstanding scaling trends, and gradually replacing convolutional neural networks (CNNs). Recent works on self-supervised learning (SSL) introduce siamese pre-training tasks, on which Transformer backbones continue to demonstrate ever stronger results than CNNs. People come to believe that Transformers or self-attention modules are inherently more suitable than CNNs in the context of SSL. However, it is noteworthy that most if not all prior arts of SSL with CNNs chose the standard ResNets as their backbones, whose architecture effectiveness is known to already lag behind advanced Vision Transformers. Therefore, it remains unclear whether the self-attention operation is crucial for the recent advances in SSL - or CNNs can deliver the same excellence with more advanced designs, too? Can we close the SSL performance gap between Transformers and CNNs? To answer these intriguing questions, we apply self-supervised pre-training to the recently proposed, stronger lager-kernel CNN architecture and conduct an apple-to-apple comparison with Transformers, in their SSL performance. Our results show that we are able to build pure CNN SSL architectures that perform on par with or better than the best SSL-trained Transformers, by just scaling up convolutional kernel sizes besides other small tweaks. Impressively, when transferring to the downstream tasks `MS COCO` detection and segmentation, our SSL pre-trained CNN model (trained in 100 epochs) achieves the same good performance as the 300-epoch pre-trained Transformer counterpart. We hope this work can help to better understand what is essential (or not) for self-supervised learning backbones. Codes will be made public.

## 1 Introduction

After leading the computer vision field for a couple of decades, the dominant position of convolutional neural networks (CNNs) (Krizhevsky et al., 2012a; Simonyan & Zisserman, 2015.; He et al., 2016; Huang et al., 2017; Howard et al., 2017; Xie et al., 2017; Tan & Le, 2019) is being vigorously challenged by recently emerging Vision Transformers (Dosovitskiy et al., 2021; Touvron et al., 2021; Wang et al., 2021; Vaswani et al., 2021; Yuan et al., 2021; Zhai et al., 2022; d'Ascoli et al., 2021; Liu et al., 2021b). The emergence of local-window self-attention (Liu et al., 2021b;a; Vaswani et al., 2021; Yang et al., 2021) significantly unleashes the power of Transformers as general backbones taking over various computer vision benchmarks rapidly with permissible resource budget, including ImageNet classification (Dosovitskiy et al., 2021), region-level object detection (Dong et al., 2022), dense pixel-level semantic segmentation (Zheng et al., 2021), and video action classification (Arnab et al., 2021).

Recently, the prominent representation power of Transformer with self-attention is also introduced into the self-supervised learning (SSL) regime. Previous art (Trinh et al., 2019) mimics the masked language modeling and conducts a preliminary exploration on masked patch prediction using ResNet (He et al., 2016). iGPT (Chen et al., 2020a) pre-trains sequence Transformers to predict pixels in an auto-regressive way as a generative model, and adopted a linear probing for classification. Dosovitskiy et al. (Dosovitskiy et al., 2021) further pre-train vanilla ViT on large-scale JFT-300M dataset, showing the promise of ViT on self-supervision. MoCo-v3 (Chen et al., 2021) generalizes contrastive learning to ViT achieving 84.1%

accuracy on ImageNet-1K (Russakovsky et al., 2015). DINO (Caron et al., 2021) proposes a self-distillation self-supervision paradigm where two ViT models fed with the same image but different views are trained to minimize their output probability distribution. They show that under this form of self-supervision, ViT explicitly contains segmentation information. Recently, EsViT (Li et al., 2021) and MoBY (Xie et al., 2021) illustrate that more advanced Swin Transformer (Liu et al., 2021b) can also be applied with SSL, standing out as a better backbone than ViT and CNNs. However, it is worth noting that most if not all previous arts chose the standard ResNets as the CNN backbone whose architectural design is known to already lag behind advanced Vision Transformers (Liu et al., 2022b), rendering unfair comparisons between Transformers and CNNs.

Since the backbone choice is a crucial ingredient to SSL, it is more desirable to draw a relatively fair comparison between CNNs and Transformers to better understand if the self-attention in Transformers is crucial to the recent advances in SSL, or if CNNs with state-of-the-art designs can have the same promise. In this paper, we turn to the recently proposed, stronger CNN architecture - ConvNeXt (Liu et al., 2022b). By modernizing the seemingly "old-fashioned" ResNet towards the design of Swin Transformer, ConvNeXt favorably rivals Swin Transformer on ImageNet classification and downstream tasks, which can therefore conduct a more apple-to-apple comparison with Transformers in the context of SSL. Our work is intended to test whether the recent strike of CNNs can be generalized to the SSL regime, as well as build a new state-of-the-art baseline for self-supervised CNNs in the era of Transformers. We briefly summarize our contributions below:

- An intriguing phenomenon is first observed in our paper: while ConvNeXt demonstrates compelling performance over strong Swin Transformers in supervised learning, its performance in SSL is no better than the original Transformer backbone - ViT.

- Nevertheless, simply adding two small adaptions (i. naively scaling the kernel size up; ii. adding Batchnorm layers after depthwise convolutions) to vanilla ConvNeXt, we are able to build attention-free CNNs, which we dubbed Big ConvNet SSL (**BC-SSL**), to perform on par or even better than the best SSL-trained Transformers with linear probe and $k$-NN evaluation on ImageNet classification, while enjoying faster inference throughput (up to 40% faster on A100 GPU).

- More impressively, when transferring to downstream tasks such as linear classification, detection, and segmentation, our modified CNN architecture demonstrates significantly larger performance gains. Simply as it is, our SSL pre-trained BC-SSL (trained in 100 epochs) achieves equally good performance to the 300-epoch pre-trained Swin Transformer counterparts on `MS COCO` object detection and segmentation.

- We also observe an encouraging trend of robustness evaluation, that is, the robustness of BC-SSL monotonously improves as the kernel size scales up to 15×15, performing an all-around win over Swin-T in terms of both clean and robust accuracy.

We mainly focus on probing self-supervised large-kernel CNNs using ConvNext in this work, yet we are aware of other CNN architectures positively equipped with even larger kernels like RepLKNet (Ding et al., 2022) and SLaK (Liu et al., 2022a), which could also be competitive baselines in this regime. Although we observe that the benefits of large kernels seem to saturate at 9×9 kernels in self-supervised ConvNeXt, we do not exclude the possibility that other architectures such as RepLKNet or SLaK can benefit more from increasing kernels further, which we leave as future work.

## 2 Related Work

### 2.1 Visual Self-Supervised Learning

Most if not all self-supervised learning methods in computer vision can be categorized as discriminative or generative (Grill et al., 2020).

Contrastive learning is a leading direction in discriminative approaches that achieve state-of-the-art SSL performance (Chen et al., 2020b; He et al., 2020; van den Oord et al., 2018; Hénaff et al., 2019; Hjelm et al., 2019; Bachman et al., 2019; He et al., 2020; Chen et al., 2020d; 2021). Contrastive methods avoid a costly pixel-level generation step and aim to learn augmentation-invariant representation by bringing the representation between different augmented pairs of the same image (positive pairs) closer and pushing the representation of augmented views from different images (negative pairs) away from each other (Wu et al., 2018; Doersch & Zisserman, 2017; Chen et al., 2020b). A drawback of this approach is the requirement of comparing features from a large number of images (including positive pairs and negative pairs) simultaneously. More importantly, such an approach usually needs a large batch of data (Chen et al., 2020b) or memory banks (He et al., 2020; Wu et al., 2018) to obtain sufficient negative pairs.

Many works start to propose various techniques to eliminate the negative pairs due to the cumbersome comparisons between different examples. DeepCluster (Caron et al., 2018) successfully avoids the usage of negative pairs by applying a clustering process. More specifically, it uses the representation from the prior phase to cluster data points, after which the cluster index of the data point is treated as the classification target for the new representation. Follow-up work continues to improve the effectiveness and efficiency of simultaneous clustering and representation learning (Asano et al., 2019; Caron et al., 2018; 2019; Huang et al., 2019; Li et al., 2020; Zhuang et al., 2019). BYOL (Grill et al., 2020) is another milestone work that effectively removes the negative pairs with strong results. BYOL feeds two networks with different augmented views of the same image. The online network is trained online to predict the representation of the target network whose weights are updated with a slow-moving average (momentum encoder) of the former. The momentum encoder was claimed to be crucial to prevent collapse. A follow-up study (Chen & He, 2021) shows that stop-gradient operation plays an essential role in preventing collapsing and BYOL works even without a momentum encoder at some performance cost. Inspired by mean teacher (Tarvainen & Valpola, 2017) and BYOL, DINO uses a self-distillation-based loss instead of a contrastive loss achieving strong SSL performance with ViT (Dosovitskiy et al., 2021). EsViT (Li et al., 2021) recently explore DINO to Swin Transformers. They propose to match the region-level features together with the view-level features for multi-stage Transformers and further establish a new state-of-the-art bar for SSL.

Generative methods seek to jointly learn data and representation together (Donahue et al., 2016; Donahue & Simonyan, 2019; Brock et al., 2018; Donahue et al., 2016) with either auto-encoding of images (Vincent et al., 2008; Kingma & Welling, 2013; Rezende et al., 2014) or adversarial learning (Goodfellow et al., 2020). Recent generative approaches revisit the mask language modeling in images as pre-training tasks have achieved competitive finetuning performance (Dosovitskiy et al., 2021; Bao et al., 2021; He et al., 2022; Zhou et al., 2021; Xie et al., 2022).

The vast majority of recent breakthroughs achieved in SSL are accompanied by advanced Transformer architecture. Therefore, it is important to decouple SSL from Transformers and to see whether self-attention-free architectures like CNN can deliver the same excellence with more advanced designs too.

## 2.2 Large Kernels in Supervised Learning

Large kernels in supervised learning have a long history, stemming from the 2010s (Krizhevsky et al., 2012b; Szegedy et al., 2015; 2017), where AlexNet (Krizhevsky et al., 2012b) adopts 11×11 kernels in the first convolutional layer for instance. Global Convolutional Network (Peng et al., 2017) replaces a 2D convolution with two parallels of stacked 1D convolution, with kernel size up to 25×1 + 1×25. The idea has recently been revisited in SegNeXt (Guo et al., 2022) to build an efficient multi-scale attention module. Perhaps predominantly due to the popularity of VGG (Simonyan & Zisserman, 2014), people start to blindly stack multiple small kernels (i.e., 1×1 or 3×3) to obtain a large receptive field for computer vision tasks (He et al., 2016; Howard et al., 2017; Xie et al., 2017; Huang et al., 2017).

Motivated by the large window size of Swin Transformer, ConvNeXt (Liu et al., 2022b) explores the inverted bottleneck design equipped with 7×7 kernels, evincing the promise of large kernels holds for CNN. RepLKNet (Ding et al., 2022) is a concurrent work that scales kernel size to 31×31 using an auxiliary 5×5 kernel. SLaK (Liu et al., 2022a) pushes the kernel size to 51×51 by employing certain decomposition and sparsity techniques, improving the training stability and memory scalability of large convolutions kernels. More

Table 1: **Linear and $k$-NN classification on ImageNet-1K.** Models are pre-trained for 100 epochs with DINO.

| Model | Kernel Size | Param (M) | Linear | $k$-NN |
|---|---|---|---|---|
| ResNet-50 | 3×3 | 23 | 67.14 | 58.88 |
| ConvNeXt-T | 3×3 | 28 | 73.27 | 67.98 |
| ConvNeXt-T | 5×5 | 28 | 73.82 | 68.34 |
| ConvNeXt-T | 7×7 | 29 | 74.10 | 68.65 |
| ViT-S | - | 21 | 73.51 | 68.77 |
| Swin-T | - | 28 | **74.98** | **69.72** |

recently, Chen et al. (2022); Xiao et al. (2022) reveals the feasibility of large kernels for 3D CNNs and time series classification too, respectively. However, the potential of modern large-kernel CNNs has never been explored in the context of self-supervised learning. Our paper conducts a pilot study asking whether we can close the SSL performance gap between Transformers and CNNs by introducing these stronger large-kernel CNN architectures.

## 3 Modern Large-Kernel CNNs in Self-Supervised Learning

In this section, we will carry out the exploration of the modern large-kernel CNNs in self-supervised learning. A brief recap of modern large-kernel CNNs is provided first, followed by the evaluation of the vanilla ConvNeXt in SSL showing inferior performance to its Transformers competitors. We consequently study several design choices built upon which we can bridge the performance gap between modern large-kernel CNNs between SoTA Transformers, i.e., Swin Transformers, in SSL.

### 3.1 A Brief Recap of ConvNeXt

ConvNeXt (Liu et al., 2022b), a recently emerging pure CNN model armed with more sophisticated architecture design, heats up the debate between CNNs and Transformers in supervised learning. It thoroughly investigates the architecture designs used in Swin Transformers and assembles a set of principles that substantially boosts the performance of a standard ResNet-50 to the level of the state-of-the-art Transformers. Specifically, ConvNeXt adopts a different stage compute ratio (1:1:9:1); a ViT-style patchify stem that divides images into non-overlapped patches; depthwise convolution and inverted bottleneck with an increase of network width; 7×7 large kernels instead of 3×3; replacing ReLU (Nair & Hinton, 2010) with GELU (Hendrycks & Gimpel, 2016); substituting BatchNorm (Ioffe, 2017) with LayerNorm (Ba et al., 2016). Built upon the following set of principles, ConvNeXt is able to match or even outperform Swin Transformers under most scenarios in supervised learning. Ding et al. (Ding et al., 2022) and Liu et al. (Liu et al., 2022a) further push along the direction of large kernel and demonstrate the possibility of scaling kernel size up to 31×31 and 51×51, while achieving even better performance, respectively.

### 3.2 Vanilla ConvNeXt in SSL

We first examine whether the vanilla ConvNeXt can close the performance gap between Transformers and CNNs. We choose the state-of-the-art self-distillation with no labels (DINO) (Caron et al., 2021) as our SSL framework with 4 architectures: ResNet-50, ConvNeXt-T, ViT-S with 16×16 patch size, and Swin-T with 4×4 patch size. We choose ConvNeXt-T as the representative of modern CNN architectures mainly due to its several similarities with the SoTA SSL-trained Swin-T (Liu et al., 2021b) in terms of (1) parameter count; (2) throughput; (3) and supervised performance. We follow the vanilla 100-epoch training configuration used in DINO and pre-train models on ImageNet-1K without labels with AdamW (Loshchilov & Hutter, 2019) and a batch size of 512. Cosine learning rate decay is used whose base learning rate is scaled with the batch size as $lr = 0.0005 * $ batchsize$/256$. Weight decay is also decayed from 0.04 to 0.4 with a cosine function. The teacher temperature of self-distillation $\tau$ is set as 0.4. We report the results of two evaluation protocols $k$-NN and linear probing in Table 1.

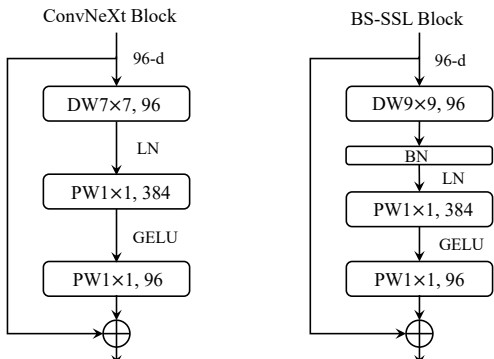

Figure 1: **Block designs for ConvNeXt and BC-SSL.** "DW" refers to depthwise convolution and "PW" stands for pointwise convolution. To fully translate the promise of modern CNNs in supervised learning to self-supervised learning, two small adaptions are adopted on the original ConvNeXt: (1) adding BatchNorm layers after large depthwise kernels; (2) naively scaling up convolutional kernel size to 9×9.

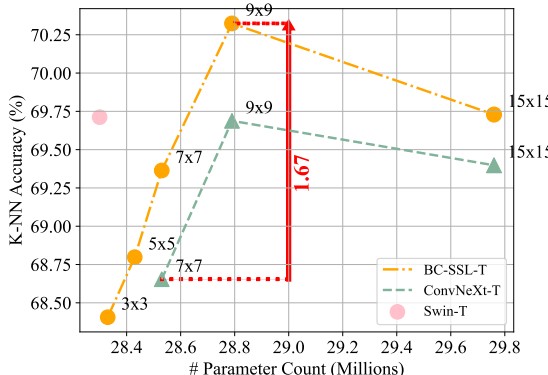

Figure 2: $k$-**NN Accuracy of ConvNeXt-T, BC-SSL-T, and Swin-T with various kernel sizes.** BC-SSL-T improves the $k$-NN accuracy over the standard ConvNeXt-T by 1.67%. Models are pre-trained for 100 epochs with DINO.

We observe that even with the standard $3 \times 3$ kernels, the superb architecture design of ConvNeXt directly brings 6.13% higher linear probing accuracy over the "old-fashion" ResNet. Increasing kernel size from 3×3 to 7×7 continuously boosts the accuracy by 0.83%. However, the performance of standard ConvNeXt with 7×7 kernels still falls behind its Transformer competitors. The above observation indicates that *the promise of modern CNN designs in supervised learning seems can not be fully translated to the SSL scenario.*

### 3.3 Pushing the Limits of ConvNeXt in SSL

**Adding BatchNorm after large depthwise kernels.** By default, ConvNeXt is a BatchNorm-free architecture. Substituting Batchnorm with LayerNorm slightly improves the performance as reported in the original work. Albeit that BatchNorm may have many intricacies that cause detrimental effects on performance (Wu & Johnson, 2021) in supervised learning, it historically plays an essential role in self-supervised learning. It has been shown that batch normalization is crucial for BYOL to achieve good performance (Grill et al., 2020; Richemond et al., 2020). MoCo-v3 (Chen et al., 2021) shows that removing all BatchNorm layers in the MLP heads causes a 2.1% accuracy drop. MoBY (Xie et al., 2021) confirms a similar phenomenon in Swin Transformers.

In contrast to these previous arts, we investigate the effect of batch normalization in the backbone. More specifically, we choose to add a BatchNorm layer after each depthwise convolutional kernel in the ConvNeXt backbone. Table 2 demonstrates that this small modification consistently improves the performance of self-supervised ConvNeXt across various batch sizes.

Table 2: **Linear classification of ConvNeXt with vs. without BatchNorm (BN) after Depthwise convolutions on ImageNet-1K.** Models are pre-trained for 100 epochs with DINO.

| Model | Linear Classification | | | | |
|---|---|---|---|---|---|
| Kernel Size | 3×3 | 5×5 | 7×7 | 9×9 | 15×15 |
| ConvNeXt-T w/o BN | 73.27 | 73.82 | 74.10 | 74.52 | 74.28 |
| ConvNeXt-T w/ BN | **74.04** | **74.35** | **74.48** | **75.01** | **74.47** |

**Naively scaling up convolutional kernel sizes.** The larger kernel benefit of ConvNeXt reaches a saturation point at 7×7 in supervised learning (Liu et al., 2022b). While it is possible to expand performance gains by further enlarging kernels, sophisticated techniques like structure re-parameterization (Ding et al., 2022) and sparsity (Liu et al., 2022a) are required. Nonetheless, we empirically find that in self-supervised learning the kernel size can be favorably scaled to 9×9 without bells and whistles as shown in Table 2. However, an accuracy drop occurs when we further increase the kernel size to 15×15. Although increasing kernel sizes beyond 9×9 does not provide more performance gains for ConvNeXt, we do not exclude the possibility that other large-kernel recipes such as RepLKNet and SLaK can benefit more from increasing kernel further, as we have observed much improvement room of promise in supervised learning (Ding et al., 2022; Liu et al., 2022a). Potential future work is how to fully explore the capacity of larger kernels beyond 9×9 in self-supervised learning.

Until now, we have finished our exploration of modern large-kernel CNNs in self-supervised learning and ended up with our modified architecture in Figure 1, which we dubbed **BC-SSL**. The above two small tweaks (adding BatchNorm after depthwise convolutions and scaling convolutional kernel sizes to 9×9) bring encouraging performance gains. Figure 2 shows that the added BatchNorm consistently brings around 0.7% $k$-NN accuracy gains to ConvNeXt across kernel sizes, and enlarging kernels from 7×7 to 9×9 further increases the performance by 0.97%, outperforming Swin Transformers. Overall we achieve an encouraging 1.67% accuracy improvement over the vanilla ConvNeXt even under a small-scale pre-training regime.

In the next section, we will evaluate the scalability of BC-SSL in terms of training time and model size, as well as the transferability on downstream tasks. *From now on, we will choose 9×9 as our default kernel size and add a BatchNorm layer after depthwise convolutions in each residual block for BC-SSL.*

## 4 Main Evaluations of BC-SSL

We first evaluate the proposed BC-SSL backbone in SSL with the standard self-supervised benchmark on ImageNet-1K (Russakovsky et al., 2015). We also evaluate the quality of the learned representations by conducting downstream transfer learning on MS COCO detection and segmentation (Lin et al., 2014), 18 small datasets, and several ImageNet-level robustness benchmarks.

### 4.1 Evaluation on ImageNet-1K

**Implementation details.** To conduct an apple-to-apple comparison between BC-SSL and Transformers, we follow the current state-of-the-art SSL-trained Transformer results (Caron et al., 2021; Li et al., 2021; Xie et al., 2021) and adopt DINO (Caron et al., 2021) as our SSL framework. We construct two variants of BC-SSL, BC-SSL-T and BC-SSL-S, to be of similar sizes to Swin-T and Swin-B. We then train them with the full 300-epoch training recipe as reported in Caron et al. (2021). Concretely, all models are trained with AdamW (Loshchilov & Hutter, 2019) and a batch size of 512, with a learning rate scaled as $lr = 0.0005 * \text{batchsize}/256$, decayed with a cosine schedule. The temperature of the teacher linearly increases from 0.04 to 0.07 within the first 30 epochs as a warm-up phase, while the temperature of the student is set to be 0.1 constantly. As suggested by Caron et al. (2021), we remove the last layer normalization of the DINO head for

Table 3: Comparison with SoTA SSL results across different architectures on ImageNet-1K. The patch size is 16×16 and 4×4 for ViT and Swin Transformers, respectively, and the window size of Swin Transformers is set as default 7×7. ViT-BN is ViT that replaces the LayerNorm before MLP blocks by BatchNorm. "RN152w2+SK" refers to ResNet-152 with 2× wider channels and selective kernels (Li et al., 2019). Throughput numbers are obtained from DINO (Caron et al., 2021) except for Swin Transformer and BC-SSL, which are measured by us using a V100 GPU (black) and an A100 GPU (blue), respectively, following (Liu et al., 2022b). On an A100 GPU, SSL pre-trained BC-SSL can have a much higher (up to 40%) throughput than SSL pre-trained Swin Transformer.

| Method | Architecture | #Parameters (M) ↓ | Throughput ↑ | Linear ↑ | $k$-NN ↑ |
|---|---|---|---|---|---|
| *SoTA SSL with Big Model Sizes* | | | | | |
| iGPT (Chen et al., 2020a) | iGPT-XL | 6801 | - | 72.0 | - |
| SCLR (Chen et al., 2020b) | RN50w4 | 375 | 117 | 76.8 | 69.3 |
| SwAV (Caron et al., 2020) | RN50w5 | 586 | 76 | 78.5 | 67.1 |
| BYOL (Grill et al., 2020) | RN50w4 | 375 | 117 | 78.6 | − |
| MoCo-v3 (Chen et al., 2021) | ViT-H-BN/16 | 632 | 32 | 79.1 | - |
| SimCLR-v2 (Chen et al., 2020c) | RN152w2+SK | 354 | - | 79.4 | - |
| BYOL (Grill et al., 2020) | RN200w2 | 250 | 123 | 79.6 | 73.9 |
| *SoTA SSL with Small Model Sizes* | | | | | |
| BYOL (Grill et al., 2020) | RN50 | 23 | 1237 | 74.4 | 64.8 |
| MoBY (Xie et al., 2021) | Swin-T | 28 | 758/1326 | 75.1 | - |
| DCv2 (Caron et al., 2020) | RN50 | 23 | 1237 | 75.2 | 67.1 |
| SwAV (Caron et al., 2020) | RN50 | 23 | 1237 | 75.3 | 65.7 |
| DINO (Caron et al., 2021) | RN50 | 23 | 1237 | 75.3 | 67.5 |
| MoCo-v3 (Chen et al., 2021) | ViT-B | 85 | 312 | 76.7 | - |
| DINO (Caron et al., 2021) | ViT-S | 21 | 1007 | 77.0 | 74.5 |
| DINO (Li et al., 2021) | Swin-T | 28 | 758/1326 | 77.0 | 74.2 |
| DINO (Ours) | BC-SSL-T | 29 | 762/1777 (+34%) | 77.8 | 75.7 |
| DINO (Caron et al., 2021) | ViT-B | 85 | 312 | 78.2 | 76.1 |
| DINO (Li et al., 2021) | Swin-S | 49 | 437/857 | 79.2 | 76.8 |
| DINO (Ours) | BC-SSL-S | 50 | 442/1197 (+40%) | 79.0 | 76.6 |

improving stability. Following BYOL (Grill et al., 2020) and DINO, we choose color jittering, Gaussian blur and solarization for the data augmentations and multi-crop (Caron et al., 2020) with a bicubic interpolation.

**Evaluation protocols.** Same as Caron et al. (2021); Li et al. (2021); Xie et al. (2021); Chen et al. (2021), we report top-1 linear probe and $k$-NN accuracy on ImageNet-1K validation set. For linear probing, random size cropping and horizontal flips are adopted as augmentation. The backbone is frozen and only the classifier is trained by SGD for 100 epochs with a small learning rate of 0.001, which is decayed with a cosine decay schedule. Besides, we also evaluate the learned representation with a simple non-parametric evaluation - $k$-Nearest neighbor (k-NN) classifiers. We first store the features of the training data learned by the pre-training backbone; find the $k$ nearest features that match the feature of a test image; and vote for the label. The comparison results are reported in Table 3.

**Comparisons with ResNets.** We first compare the performance of BC-SSL with the standard ResNet-50 (23M) mainly shown on the bottom panel. We observe that BC-SSL-T dramatically outperforms the best SSL-trained ResNet-50 (DINO) by 2.5% with linear probe and 8.2% with $k$-NN, demonstrating that the superiority of modern CNNs over the conventional ResNets can be generalized to the longer training time regime. When compared with bigger ResNets on the top panel, our 300-epoch trained BC-SSL-S with 50M parameters is able to achieve comparable performance to those large-scale ResNets with hundreds of parameters (usually trained for around 1000 epochs), while enjoying up to 5.8× higher throughput.

**Comparisons with Transformers.** We next compare BC-SSL with two strong Transformer baselines: ViT and Swin Transformer as reported on the bottom panel in Table 3. Overall, BC-SSL provides a positive signal that modern CNNs perform satisfactorily against two strong Transformer baselines in terms of the accuracy-computation trade-off. Without using any sophisticated attention modules, BC-SSL-T outperforms

Swin-T by 0.8% with linear probing and 1.5% with $k$-NN. Our larger model BC-SSL-S further boosts the linear classification accuracy to 79.0%, matching the performance of Swin-S, while being slightly faster in inference throughput than the latter when tested on a V100 GPU (442 vs 437). However, when being benchmarked on the more advanced A100 GPU marked with blue colors, BC-SSL-S is 40% faster than Swin-S (1197 vs 857), thanks to the efficient convolutional modules and simple design choices. Moreover, the promise of BC-SSL also holds when compared to the Transformers trained with contrastive methods, i.e., ViT-S trained with MoCo-v3, achieving 2.3% higher accuracy.

## 4.2 Evaluation on Downstream COCO Object Detection and Segmentation

One advantage of convolutional architectures compared with Transformers with global self-attention (Vaswani et al., 2017; Dosovitskiy et al., 2021) is its preferable low computational complexity with respect to image size, allowing it to be efficiently transferred to downstream tasks with high resolution. To evaluate the transferability of the learned representations of BC-SSL, we also conduct transfer learning on MS COCO object detection and segmentation (Lin et al., 2014). Following previous arts (Li et al., 2021; Xie et al., 2021; Liu et al., 2022b), We finetune Mask R-CNN (He et al., 2017) on the COCO dataset with BC-SSL backbone for a 3× (36 epochs) schedule. Layer-wise learning rate decay (Bao et al., 2021) and stochastic depth rate are adopted. The hyperparameters are exactly the same as the ones reported in supervised ConvNeXt (Liu et al., 2022b). By using this set of hyperparameters and configurations, we not only can conduct a fair comparison among various SSL results, but also can compare our self-supervised models with their supervised counterparts. To better understand the behavior of BC-SSL in different SSL training regimes, we evaluate two groups of BC-SSL models: 100-epoch pre-trained models and 300-epoch pre-trained models. Table 4 shows the results. We summarize the main observations below:

Table 4: **Object detection and segmentation on MS COCO.** Models are pre-trained on ImageNet-1K and finetuned using Mask-RCNN for 36 epochs. The performance of BC-SSL is implemented by us and the results of other models are obtained from their original papers.

| Method | Architecture | Kernel Size | $AP^{box}$ ↑ | $AP^{box}_{50}$ ↑ | $AP^{box}_{75}$ ↑ | $AP^{mask}$ ↑ | $AP^{mask}_{50}$ ↑ | $AP^{mask}_{75}$ ↑ |
|---|---|---|---|---|---|---|---|---|
| *300-epoch supervised learning* | | | | | | | | |
| Supervised (Liu et al., 2022b) | Swin-T | - | 46.0 | 68.1 | 50.3 | 41.6 | 65.1 | 44.9 |
| Supervised (Liu et al., 2022b) | ConvNeXt-T | 7×7 | 46.2 | 67.9 | 50.8 | 41.7 | 65.0 | 44.9 |
| *100-epoch SSL pre-training* | | | | | | | | |
| DINO | BC-SSL-T | 3×3 | 45.0 | 66.3 | 49.6 | 40.5 | 63.5 | 43.6 |
| DINO | BC-SSL-T | 5×5 | 45.9 | 67.3 | 50.6 | 41.1 | 64.3 | 44.3 |
| DINO | BC-SSL-T | 7×7 | 46.1 | 67.4 | 50.8 | 41.2 | 64.5 | 44.2 |
| DINO | BC-SSL-T | 9×9 | **46.3** | **67.6** | **51.0** | **41.3** | **64.7** | **44.4** |
| *300-epoch SSL pre-training* | | | | | | | | |
| DINO (Li et al., 2021) | Swin-T | - | 46.2 | 67.9 | 50.5 | **41.7** | 64.8 | **45.1** |
| EsViT (Li et al., 2021) | Swin-T | - | 46.2 | 68.0 | 50.6 | 41.6 | 64.9 | 44.8 |
| DINO | BC-SSL-T | 9×9 | **46.6** | **68.1** | **51.3** | 41.6 | **65.0** | 44.5 |

① **Performance increases as the kernel size.** In the middle group of Table 4, we report the performance of BC-SSL (trained in 100 epochs) with increasing kernel sizes from 3×3 to 9×9. We can observe a very clear trend that the performance increases as the kernel size. While BC-SSL-T with 3×3 kernels suffers from a big performance gap to 300-epoch pre-trained Swin-T, it gradually approaches and eventually matches the performance of the latter using 9×9 kernels. The performance makes sense since larger kernels obtain a larger effective receptive field (ERF) and benefit more on the high-resolution dense prediction tasks (Liu et al., 2022a). This result highlights the better transferability of large-kernel convolutions over the Transformers on dense prediction downstream tasks.

② **BC-SSL outperforms self-supervised Transformers.** When pre-trained with the full training recipe in 300 epochs, BC-SSL-T further boosts the performance of self-supervised CCNs on COCO over the Swin-T by a good margin, especially in terms of the box AP.

③ **BC-SSL performs better than its supervised counterpart.** Moreover, our self-supervised BC-SSL also outperforms its supervised counterpart as reported on the top panel. Given the accuracy of self-supervised BC-SSL falls short of the supervised one reported in (Liu et al., 2022b), this phenomenon indicates that the large kernel design in SSL brings more benefits to downstream tasks than the pre-training ImageNet task.

### 4.3 Evaluation on Robustness

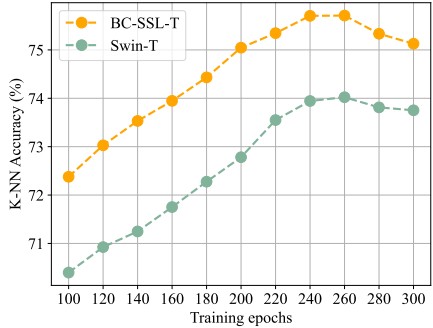

Figure 3: **Training curves of different architectures using DINO.**

Recent studies on out-of-distribution robustness (Bai et al., 2021; Paul & Chen, 2022; Zhang et al., 2022; Mao et al., 2022) show that Transformers are much more robust than CNNs when testing under distribution shifts. For instance, Mao et al. (Mao et al., 2022) demonstrate that DeiT (Touvron et al., 2021), Swin-T, and RVI (Mao et al., 2022) achieve stronger robustness than ResNet. Given the advanced architecture designs in BC-SSL, we ask if the modern large-kernel CNNs can launch a successful counterattack in self-supervised learning?

To answer this question, we directly test our linear probing classification models on several robustness benchmarks including ImageNet-A (Hendrycks et al., 2021b), ImageNet-R (Hendrycks et al., 2021a), ImageNet-Sketch (Wang et al., 2019) and ImageNet-C (Hendrycks & Dietterich, 2019) datasets. Mean corruption error (mCE) is reported for ImageNet-C and top-1 accuracy is reported for the rest of datasets.

We again observe an encouraging trend in Table 5, that is, the robustness of BC-SSL monotonously improves as the kernel size scales up to 15×15. First, Swin-T indeed achieves better robustness performance than ResNet-50 in both settings, confirming the findings in (Bai et al., 2021; Mao et al., 2022). It is then interesting to observe that BC-SSL with the smallest 3×3 kernels is already more robust than Swin-T, and our 9×9 model further performs an all-around win over Swin-T including both clean and robust accuracy, being blessed by large kernels. It is worth noting that while the 15×15 kernel undergoes a small clean accuracy drop compared to 9×9 kernel, it brings a notable improvement in robustness demonstrating the promise of large kernels in the context of robustness.

Table 5: **Robustness evaluation of BC-SSL.** All results are obtained by directly testing our ImageNet-1K linear probing models on several robustness benchmark datasets. We do not make use of any specialized modules or additional fine-tuning procedures.

| Method | Architecture | Kernel Size | FLOPs (G) / #Param. (M) ↓ | Clean (linear) ↑ | Clean ($k$-NN) ↑ | C ↓ | SK ↑ | R ↑ | A ↑ |
|---|---|---|---|---|---|---|---|---|---|
| *100-epoch SSL pre-training* | | | | | | | | | |
| DINO | ResNet-50 | 3×3 | 4.1 / 23 | 67.1 | 58.9 | 71.85 | 14.77 | 15.02 | 0.76 |
| DINO | ViT-S | - | 4.6 / 21 | 73.5 | 68.8 | 59.57 | 20.08 | 21.89 | 3.04 |
| DINO | Swin-T | - | 4.5 / 28 | 75.0 | 69.7 | 61.86 | 20.16 | 20.18 | 3.49 |
| DINO | BC-SSL-T | 3×3 | 4.4 / 28 | 74.0 | 68.4 | 61.82 | 23.18 | 21.60 | 3.80 |
| DINO | BC-SSL-T | 5×5 | 4.4 / 28 | 74.4 | 68.8 | 61.48 | 23.45 | 22.92 | 4.75 |
| DINO | BC-SSL-T | 7×7 | 4.5 / 29 | 74.5 | 69.4 | 60.40 | 24.55 | 22.92 | 4.74 |
| DINO | BC-SSL-T | 9×9 | / 29 | **75.0** | **70.3** | 59.39 | 24.55 | 22.79 | 4.66 |
| DINO | BC-SSL-T | 15×15 | 4.8 / 30 | 74.5 | 69.7 | **58.80** | **25.33** | **23.25** | **5.31** |
| *300-epoch SSL pre-training* | | | | | | | | | |
| DINO | ResNet-50 | 3×3 | 4.1 / 23 | 75.3 | 67.5 | 70.28 | 14.24 | 14.4 | 0.88 |
| DINO | Swin-T | - | 4.5 / 28 | 77.0 | 74.2 | 59.39 | 21.96 | 21.18 | 5.34 |
| DINO | BC-SSL-T | 9×9 | 4.5 / 29 | **77.8** | **75.7** | **57.44** | **25.32** | **23.82** | **5.72** |

## 5 Qualitative Study

### 5.1 $k$-NN Monitor

$k$-NN Monitor is a widely used tool to monitor training dynamics of self-supervised learning (Wu et al., 2018; Chen et al., 2021). To better understand the training dynamics of different architectures, we sparsely perform $k$-NN evaluation every 20 epochs and depict the results in Figure 3. BC-SSL-T shares an extremely similar pattern with Swin-T albeit with a consistently higher accuracy: an upsurging increase followed by a slight drop. The peak accuracy is reached around the 260 epoch.

### 5.2 Visualization

In this section, we provide visualizations through two popular tools Grad-CAM (Selvaraju et al., 2017) and Eigen-CAM (Muhammad & Yeasin, 2020) to understand the mechanism discrepancy behind the decisions made by different SSL-trained architectures. Grad-CAM is a label-dependent localization technique that can generate visual explanations for various architectures including CNNs and Transformers. Eigen-CAM is a label-free technique that takes the first principle component of the 2D activations to generate bounding boxes for object localization and segmentation. This combination can evaluate the quality of both, the representations learned by backbone and linear probing. Following the tutorials in (Gildenblat & contributors, 2021), we choose the representation learned by the last layer from the last stage to visualize CNN models. For Transformers, we choose the features after the first LayerNorm layer in the last stage's last block. Since the activation of Transformers is usually not 2D, we further reshape it to 2D spatial images.

We compare across ResNet-50, ViT-S, Swin-T, and BC-SSL-T with various kernel sizes in Appendix Figure 4. From the heatmaps, we can conclude that CNNs with 3×3 kernels (ResNet-50 and BC-SSL) either capture the smallest range of important pixels or no important pixels to make decisions. As the kernel size continuously increases, the red regions (corresponding to high scores) also gradually expand, and desirably cover the labeled object when the kernels are large enough, i.e., 9×9, 15×15. This indicates that large kernels inherently have a larger effective receptive field than smaller kernels, leading to more robust and accurate prediction. On the other hand, it seems that Transformers with self-attention tend to be good at capturing shapes than CNNs, although sometimes their red, shaped heatmaps are completely located on the background. This behavior is in line with the findings in supervised learning (Diesendruck & Bloom, 2003), where ViT are reported to have a higher shape bias than CNNs, whereas CNNs usually tend to preserve textures rather than shapes.

## 6 Conclusions

This work does not propose a novel method or model but instead provides an empirical study on an incremental baseline inspired by the recent breakthroughs in self-supervised learning: rethinking self-supervised CNNs in the era of Transformers. Increasingly stronger results achieved by new self-supervised training recipes accompanied by advanced Transformers make people start to believe that Transformers or self-attention operations are inherently more suitable than CNNs in the context of SSL. In this paper, we decouple SSL from Transformers and ask whether self-attention-free architectures like CNN can deliver the same excellence with more advanced designs too. We provide an encouraging signal that we are able to build pure CNN SSL architectures that perform on par with or better than the best SSL-trained Transformers, by just scaling up convolutional kernel sizes besides several small tweaks. Our results highlight that the simple design of convolutional operations remains powerful in self-supervised learning.

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

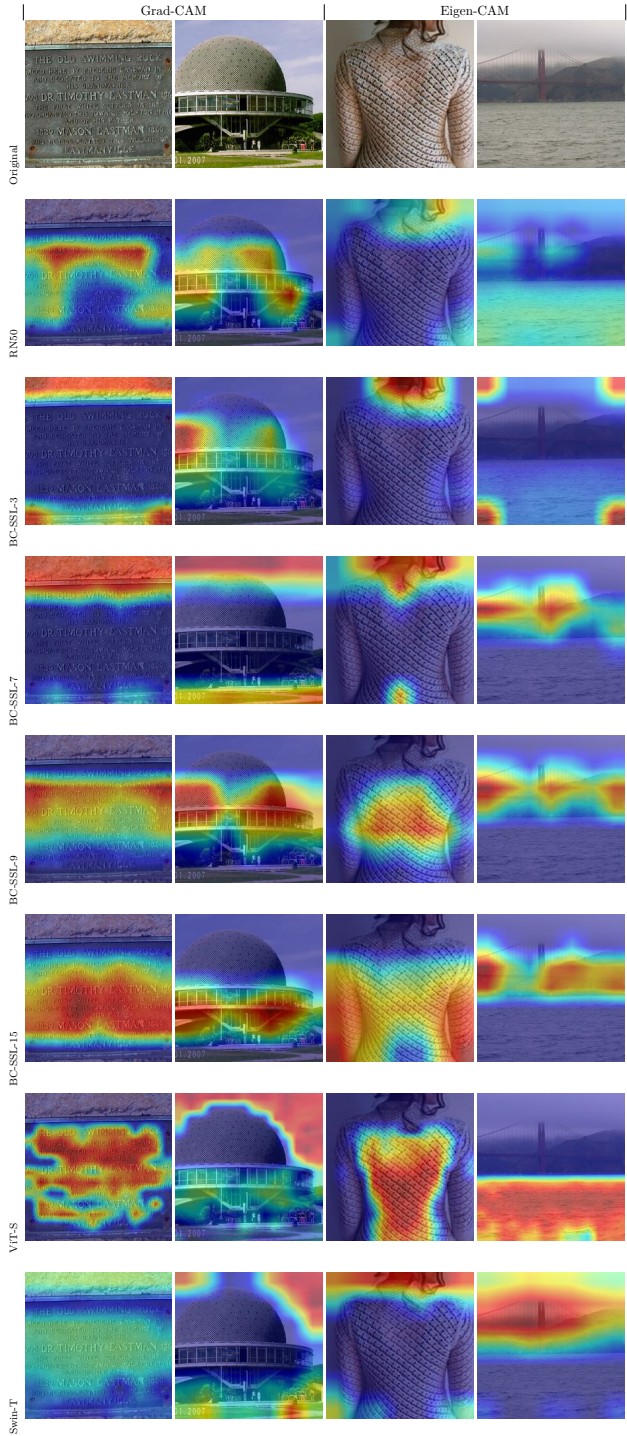

Figure 4: Heatmap visualization generated by different models.

Jimmy Lei Ba, Jamie Ryan Kiros, and Geoffrey E Hinton. Layer normalization. *arXiv preprint arXiv:1607.06450*, 2016.

Philip Bachman, R Devon Hjelm, and William Buchwalter. Learning representations by maximizing mutual information across views. In *Neural Information Processing Systems*, 2019.

Yutong Bai, Jieru Mei, Alan L Yuille, and Cihang Xie. Are transformers more robust than cnns? *Advances in Neural Information Processing Systems*, 34:26831–26843, 2021.

Hangbo Bao, Li Dong, and Furu Wei. Beit: Bert pre-training of image transformers. *arXiv preprint arXiv:2106.08254*, 2021.

Andrew Brock, Jeff Donahue, and Karen Simonyan. Large scale gan training for high fidelity natural image synthesis. *arXiv preprint arXiv:1809.11096*, 2018.

Mathilde Caron, Piotr Bojanowski, Armand Joulin, and Matthijs Douze. Deep clustering for unsupervised learning of visual features. In *Proceedings of the European conference on computer vision (ECCV)*, pp. 132–149, 2018.

Mathilde Caron, Piotr Bojanowski, Julien Mairal, and Armand Joulin. Unsupervised pre-training of image features on non-curated data. In *Proceedings of the IEEE/CVF International Conference on Computer Vision*, pp. 2959–2968, 2019.

Mathilde Caron, Ishan Misra, Julien Mairal, Priya Goyal, Piotr Bojanowski, and Armand Joulin. Unsupervised learning of visual features by contrasting cluster assignments. *Advances in Neural Information Processing Systems*, 33:9912–9924, 2020.

Mathilde Caron, Hugo Touvron, Ishan Misra, Hervé Jégou, Julien Mairal, Piotr Bojanowski, and Armand Joulin. Emerging properties in self-supervised vision transformers. In *Proceedings of the IEEE/CVF International Conference on Computer Vision*, pp. 9650–9660, 2021.

Mark Chen, Alec Radford, Rewon Child, Jeffrey Wu, Heewoo Jun, David Luan, and Ilya Sutskever. Generative pretraining from pixels. In *International conference on machine learning*, pp. 1691–1703. PMLR, 2020a.

Ting Chen, Simon Kornblith, Mohammad Norouzi, and Geoffrey Hinton. A simple framework for contrastive learning of visual representations. In Hal Daumé III and Aarti Singh (eds.), *Proceedings of the 37th International Conference on Machine Learning*, volume 119 of *Proceedings of Machine Learning Research*, pp. 1597–1607. PMLR, 13–18 Jul 2020b. URL https://proceedings.mlr.press/v119/chen20j.html.

Ting Chen, Simon Kornblith, Kevin Swersky, Mohammad Norouzi, and Geoffrey E Hinton. Big self-supervised models are strong semi-supervised learners. *Advances in neural information processing systems*, 33:22243–22255, 2020c.

Xinlei Chen and Kaiming He. Exploring simple siamese representation learning. In *Proceedings of the IEEE/CVF Conference on Computer Vision and Pattern Recognition*, pp. 15750–15758, 2021.

Xinlei Chen, Haoqi Fan, Ross Girshick, and Kaiming He. Improved baselines with momentum contrastive learning. *arXiv preprint arXiv:2003.04297*, 2020d.

Xinlei Chen, Saining Xie, and Kaiming He. An empirical study of training self-supervised vision transformers. In *Proceedings of the IEEE/CVF International Conference on Computer Vision*, pp. 9640–9649, 2021.

Yukang Chen, Jianhui Liu, Xiaojuan Qi, Xiangyu Zhang, Jian Sun, and Jiaya Jia. Scaling up kernels in 3d cnns. *arXiv preprint arXiv:2206.10555*, 2022.

Gil Diesendruck and Paul Bloom. How specific is the shape bias? *Child development*, 74(1):168–178, 2003.

Xiaohan Ding, Xiangyu Zhang, Yizhuang Zhou, Jungong Han, Guiguang Ding, and Jian Sun. Scaling up your kernels to 31x31: Revisiting large kernel design in cnns. *arXiv preprint arXiv:2203.06717*, 2022.

Carl Doersch and Andrew Zisserman. Multi-task self-supervised visual learning. In *Proceedings of the IEEE international conference on computer vision*, pp. 2051–2060, 2017.

Jeff Donahue and Karen Simonyan. Large scale adversarial representation learning. *Advances in neural information processing systems*, 32, 2019.

Jeff Donahue, Philipp Krähenbühl, and Trevor Darrell. Adversarial feature learning. *arXiv preprint arXiv:1605.09782*, 2016.

Xiaoyi Dong, Jianmin Bao, Dongdong Chen, Weiming Zhang, Nenghai Yu, Lu Yuan, Dong Chen, and Baining Guo. Cswin transformer: A general vision transformer backbone with cross-shaped windows. In *Proceedings of the IEEE/CVF Conference on Computer Vision and Pattern Recognition*, pp. 12124–12134, 2022.

Alexey Dosovitskiy, Lucas Beyer, Alexander Kolesnikov, Dirk Weissenborn, Xiaohua Zhai, Thomas Unterthiner, Mostafa Dehghani, Matthias Minderer, Georg Heigold, Sylvain Gelly, Jakob Uszkoreit, and Neil Houlsby. An image is worth 16x16 words: Transformers for image recognition at scale. In *International Conference on Learning Representations*, 2021. URL `https://openreview.net/forum?id=YicbFdNTTy`.

Stéphane d'Ascoli, Hugo Touvron, Matthew L Leavitt, Ari S Morcos, Giulio Biroli, and Levent Sagun. Convit: Improving vision transformers with soft convolutional inductive biases. In *International Conference on Machine Learning*, pp. 2286–2296. PMLR, 2021.

Jacob Gildenblat and contributors. Pytorch library for cam methods. `https://github.com/jacobgil/pytorch-grad-cam`, 2021.

Ian Goodfellow, Jean Pouget-Abadie, Mehdi Mirza, Bing Xu, David Warde-Farley, Sherjil Ozair, Aaron Courville, and Yoshua Bengio. Generative adversarial networks. *Communications of the ACM*, 63(11): 139–144, 2020.

Jean-Bastien Grill, Florian Strub, Florent Altché, Corentin Tallec, Pierre Richemond, Elena Buchatskaya, Carl Doersch, Bernardo Avila Pires, Zhaohan Guo, Mohammad Gheshlaghi Azar, et al. Bootstrap your own latent-a new approach to self-supervised learning. *Advances in neural information processing systems*, 33:21271–21284, 2020.

Meng-Hao Guo, Cheng-Ze Lu, Qibin Hou, Zhengning Liu, Ming-Ming Cheng, and Shi-Min Hu. Segnext: Rethinking convolutional attention design for semantic segmentation. *arXiv preprint arXiv:2209.08575*, 2022.

Kaiming He, Xiangyu Zhang, Shaoqing Ren, and Jian Sun. Deep residual learning for image recognition. In *2016 IEEE Conference on Computer Vision and Pattern Recognition (CVPR)*, pp. 770–778, 2016. doi: 10.1109/CVPR.2016.90.

Kaiming He, Georgia Gkioxari, Piotr Dollár, and Ross Girshick. Mask r-cnn. In *Proceedings of the IEEE international conference on computer vision*, pp. 2961–2969, 2017.

Kaiming He, Haoqi Fan, Yuxin Wu, Saining Xie, and Ross Girshick. Momentum contrast for unsupervised visual representation learning. In *Proceedings of the IEEE/CVF conference on computer vision and pattern recognition*, pp. 9729–9738, 2020.

Kaiming He, Xinlei Chen, Saining Xie, Yanghao Li, Piotr Dollár, and Ross Girshick. Masked autoencoders are scalable vision learners. In *Proceedings of the IEEE/CVF Conference on Computer Vision and Pattern Recognition*, pp. 16000–16009, 2022.

Olivier J. Hénaff, Aravind Srinivas, Jeffrey De Fauw, Ali Razavi, Carl Doersch, S. M. Ali Eslami, and Aäron van den Oord. Data-efficient image recognition with contrastive predictive coding. In *International Conference on Machine Learning*, 2019.

Dan Hendrycks and Thomas Dietterich. Benchmarking neural network robustness to common corruptions and perturbations. *arXiv preprint arXiv:1903.12261*, 2019.

Dan Hendrycks and Kevin Gimpel. Gaussian error linear units (gelus). *arXiv preprint arXiv:1606.08415*, 2016.

Dan Hendrycks, Steven Basart, Norman Mu, Saurav Kadavath, Frank Wang, Evan Dorundo, Rahul Desai, Tyler Zhu, Samyak Parajuli, Mike Guo, et al. The many faces of robustness: A critical analysis of out-of-distribution generalization. In *Proceedings of the IEEE/CVF International Conference on Computer Vision*, pp. 8340–8349, 2021a.

Dan Hendrycks, Kevin Zhao, Steven Basart, Jacob Steinhardt, and Dawn Song. Natural adversarial examples. In *Proceedings of the IEEE/CVF Conference on Computer Vision and Pattern Recognition*, pp. 15262–15271, 2021b.

R Devon Hjelm, Alex Fedorov, Samuel Lavoie-Marchildon, Karan Grewal, Adam Trischler, and Yoshua Bengio. Learning deep representations by mutual information estimation and maximization. *arXiv preprint arXiv:1808.06670*, 2019.

Andrew G Howard, Menglong Zhu, Bo Chen, Dmitry Kalenichenko, Weijun Wang, Tobias Weyand, Marco Andreetto, and Hartwig Adam. Mobilenets: Efficient convolutional neural networks for mobile vision applications. *arXiv preprint arXiv:1704.04861*, 2017.

Gao Huang, Zhuang Liu, Laurens Van Der Maaten, and Kilian Q Weinberger. Densely connected convolutional networks. In *Proceedings of the IEEE conference on computer vision and pattern recognition*, pp. 4700–4708, 2017.

Jiabo Huang, Qi Dong, Shaogang Gong, and Xiatian Zhu. Unsupervised deep learning by neighbourhood discovery. In *International Conference on Machine Learning*, pp. 2849–2858. PMLR, 2019.

Sergey Ioffe. Batch renormalization: Towards reducing minibatch dependence in batch-normalized models. *Advances in neural information processing systems*, 30, 2017.

Diederik P Kingma and Max Welling. Auto-encoding variational bayes. *arXiv preprint arXiv:1312.6114*, 2013.

Alex Krizhevsky, Ilya Sutskever, and Geoffrey E Hinton. Imagenet classification with deep convolutional neural networks. In F. Pereira, C. J. C. Burges, L. Bottou, and K. Q. Weinberger (eds.), *Advances in Neural Information Processing Systems*, volume 25. Curran Associates, Inc., 2012a. URL `https://proceedings.neurips.cc/paper/2012/file/c399862d3b9d6b76c8436e924a68c45b-Paper.pdf`.

Alex Krizhevsky, Ilya Sutskever, and Geoffrey E Hinton. Imagenet classification with deep convolutional neural networks. *Advances in neural information processing systems*, 25, 2012b.

Chunyuan Li, Jianwei Yang, Pengchuan Zhang, Mei Gao, Bin Xiao, Xiyang Dai, Lu Yuan, and Jianfeng Gao. Efficient self-supervised vision transformers for representation learning. *arXiv preprint arXiv:2106.09785*, 2021.

Junnan Li, Pan Zhou, Caiming Xiong, and Steven CH Hoi. Prototypical contrastive learning of unsupervised representations. *arXiv preprint arXiv:2005.04966*, 2020.

Xiang Li, Wenhai Wang, Xiaolin Hu, and Jian Yang. Selective kernel networks. In *Proceedings of the IEEE/CVF conference on computer vision and pattern recognition*, pp. 510–519, 2019.

Tsung-Yi Lin, Michael Maire, Serge Belongie, James Hays, Pietro Perona, Deva Ramanan, Piotr Dollár, and C Lawrence Zitnick. Microsoft coco: Common objects in context. In *European conference on computer vision*, pp. 740–755. Springer, 2014.

Shiwei Liu, Tianlong Chen, Xiaohan Chen, Xuxi Chen, Qiao Xiao, Boqian Wu, Mykola Pechenizkiy, Decebal Mocanu, and Zhangyang Wang. More convnets in the 2020s: Scaling up kernels beyond 51x51 using sparsity. *arXiv preprint arXiv:2207.03620*, 2022a.

Ze Liu, Han Hu, Yutong Lin, Zhuliang Yao, Zhenda Xie, Yixuan Wei, Jia Ning, Yue Cao, Zheng Zhang, Li Dong, et al. Swin transformer v2: Scaling up capacity and resolution. *arXiv preprint arXiv:2111.09883*, 2021a.

Ze Liu, Yutong Lin, Yue Cao, Han Hu, Yixuan Wei, Zheng Zhang, Stephen Lin, and Baining Guo. Swin transformer: Hierarchical vision transformer using shifted windows. In *Proceedings of the IEEE/CVF International Conference on Computer Vision*, pp. 10012–10022, 2021b.

Zhuang Liu, Hanzi Mao, Chao-Yuan Wu, Christoph Feichtenhofer, Trevor Darrell, and Saining Xie. A convnet for the 2020s. *arXiv preprint arXiv:2201.03545*, 2022b.

Ilya Loshchilov and Frank Hutter. Decoupled weight decay regularization. In *International Conference on Learning Representations*, 2019. URL https://openreview.net/forum?id=Bkg6RiCqY7.

Xiaofeng Mao, Gege Qi, Yuefeng Chen, Xiaodan Li, Ranjie Duan, Shaokai Ye, Yuan He, and Hui Xue. Towards robust vision transformer. In *Proceedings of the IEEE/CVF Conference on Computer Vision and Pattern Recognition*, pp. 12042–12051, 2022.

Mohammed Bany Muhammad and Mohammed Yeasin. Eigen-cam: Class activation map using principal components. In *2020 International Joint Conference on Neural Networks (IJCNN)*, pp. 1–7. IEEE, 2020.

Vinod Nair and Geoffrey E Hinton. Rectified linear units improve restricted boltzmann machines. In *Icml*, 2010.

Sayak Paul and Pin-Yu Chen. Vision transformers are robust learners. In *Proceedings of the AAAI Conference on Artificial Intelligence*, volume 36, pp. 2071–2081, 2022.

Chao Peng, Xiangyu Zhang, Gang Yu, Guiming Luo, and Jian Sun. Large kernel matters–improve semantic segmentation by global convolutional network. In *Proceedings of the IEEE conference on computer vision and pattern recognition*, pp. 4353–4361, 2017.

Danilo Jimenez Rezende, Shakir Mohamed, and Daan Wierstra. Stochastic backpropagation and variational inference in deep latent gaussian models. In *International conference on machine learning*, volume 2, pp. 2, 2014.

Pierre H Richemond, Jean-Bastien Grill, Florent Altché, Corentin Tallec, Florian Strub, Andrew Brock, Samuel Smith, Soham De, Razvan Pascanu, Bilal Piot, et al. Byol works even without batch statistics. *arXiv preprint arXiv:2010.10241*, 2020.

Olga Russakovsky, Jia Deng, Hao Su, Jonathan Krause, Sanjeev Satheesh, Sean Ma, Zhiheng Huang, Andrej Karpathy, Aditya Khosla, Michael Bernstein, et al. Imagenet large scale visual recognition challenge. *International journal of computer vision*, 115(3):211–252, 2015.

Ramprasaath R Selvaraju, Michael Cogswell, Abhishek Das, Ramakrishna Vedantam, Devi Parikh, and Dhruv Batra. Grad-cam: Visual explanations from deep networks via gradient-based localization. In *Proceedings of the IEEE international conference on computer vision*, pp. 618–626, 2017.

K. Simonyan and A. Zisserman. Very deep convolutional networks for large-scale image recognition. In *International Conference on Learning Representations*, 2015.

Karen Simonyan and Andrew Zisserman. Very deep convolutional networks for large-scale image recognition. *arXiv preprint arXiv:1409.1556*, 2014.

Christian Szegedy, Wei Liu, Yangqing Jia, Pierre Sermanet, Scott Reed, Dragomir Anguelov, Dumitru Erhan, Vincent Vanhoucke, and Andrew Rabinovich. Going deeper with convolutions. In *Proceedings of the IEEE conference on computer vision and pattern recognition*, pp. 1–9, 2015.

Christian Szegedy, Sergey Ioffe, Vincent Vanhoucke, and Alexander A Alemi. Inception-v4, inception-resnet and the impact of residual connections on learning. In *Thirty-first AAAI conference on artificial intelligence*, 2017.

Mingxing Tan and Quoc Le. EfficientNet: Rethinking model scaling for convolutional neural networks. In Kamalika Chaudhuri and Ruslan Salakhutdinov (eds.), *Proceedings of the 36th International Conference on Machine Learning*, volume 97 of *Proceedings of Machine Learning Research*, pp. 6105–6114. PMLR, 09–15 Jun 2019.

Antti Tarvainen and Harri Valpola. Mean teachers are better role models: Weight-averaged consistency targets improve semi-supervised deep learning results. *Advances in neural information processing systems*, 30, 2017.

Hugo Touvron, Matthieu Cord, Matthijs Douze, Francisco Massa, Alexandre Sablayrolles, and Hervé Jégou. Training data-efficient image transformers & distillation through attention. In *International Conference on Machine Learning*, pp. 10347–10357. PMLR, 2021.

Trieu H Trinh, Minh-Thang Luong, and Quoc V Le. Selfie: Self-supervised pretraining for image embedding. *arXiv preprint arXiv:1906.02940*, 2019.

Aäron van den Oord, Yazhe Li, and Oriol Vinyals. Representation learning with contrastive predictive coding. *arXiv preprint arXiv:1807.03748*, 2018.

Ashish Vaswani, Noam Shazeer, Niki Parmar, Jakob Uszkoreit, Llion Jones, Aidan N Gomez, Łukasz Kaiser, and Illia Polosukhin. Attention is all you need. *Advances in neural information processing systems*, 30, 2017.

Ashish Vaswani, Prajit Ramachandran, Aravind Srinivas, Niki Parmar, Blake Hechtman, and Jonathon Shlens. Scaling local self-attention for parameter efficient visual backbones. In *Proceedings of the IEEE/CVF Conference on Computer Vision and Pattern Recognition*, pp. 12894–12904, 2021.

Pascal Vincent, Hugo Larochelle, Yoshua Bengio, and Pierre-Antoine Manzagol. Extracting and composing robust features with denoising autoencoders. In *Proceedings of the 25th international conference on Machine learning*, pp. 1096–1103, 2008.

Haohan Wang, Songwei Ge, Zachary Lipton, and Eric P Xing. Learning robust global representations by penalizing local predictive power. *Advances in Neural Information Processing Systems*, 32, 2019.

Wenhai Wang, Enze Xie, Xiang Li, Deng-Ping Fan, Kaitao Song, Ding Liang, Tong Lu, Ping Luo, and Ling Shao. Pyramid vision transformer: A versatile backbone for dense prediction without convolutions. In *Proceedings of the IEEE/CVF International Conference on Computer Vision*, pp. 568–578, 2021.

Yuxin Wu and Justin Johnson. Rethinking" batch" in batchnorm. *arXiv preprint arXiv:2105.07576*, 2021.

Zhirong Wu, Yuanjun Xiong, Stella X Yu, and Dahua Lin. Unsupervised feature learning via non-parametric instance discrimination. In *Proceedings of the IEEE conference on computer vision and pattern recognition*, pp. 3733–3742, 2018.

Qiao Xiao, Boqian Wu, Yu Zhang, Shiwei Liu, Mykola Pechenizkiy, Elena Mocanu, and Decebal Constantin Mocanu. Dynamic sparse network for time series classification: Learning what to "see". In Alice H. Oh, Alekh Agarwal, Danielle Belgrave, and Kyunghyun Cho (eds.), *Advances in Neural Information Processing Systems*, 2022. URL https://openreview.net/forum?id=ZxOO5jfqSYw.

Saining Xie, Ross Girshick, Piotr Dollár, Zhuowen Tu, and Kaiming He. Aggregated residual transformations for deep neural networks. In *Proceedings of the IEEE conference on computer vision and pattern recognition*, pp. 1492–1500, 2017.

Zhenda Xie, Yutong Lin, Zhuliang Yao, Zheng Zhang, Qi Dai, Yue Cao, and Han Hu. Self-supervised learning with swin transformers. *arXiv preprint arXiv:2105.04553*, 2021.

Zhenda Xie, Zheng Zhang, Yue Cao, Yutong Lin, Jianmin Bao, Zhuliang Yao, Qi Dai, and Han Hu. Simmim: A simple framework for masked image modeling. In *Proceedings of the IEEE/CVF Conference on Computer Vision and Pattern Recognition*, pp. 9653–9663, 2022.

Jianwei Yang, Chunyuan Li, Pengchuan Zhang, Xiyang Dai, Bin Xiao, Lu Yuan, and Jianfeng Gao. Focal self-attention for local-global interactions in vision transformers. *arXiv preprint arXiv:2107.00641*, 2021.

Li Yuan, Yunpeng Chen, Tao Wang, Weihao Yu, Yujun Shi, Zi-Hang Jiang, Francis EH Tay, Jiashi Feng, and Shuicheng Yan. Tokens-to-token vit: Training vision transformers from scratch on imagenet. In *Proceedings of the IEEE/CVF International Conference on Computer Vision*, pp. 558–567, 2021.

Xiaohua Zhai, Alexander Kolesnikov, Neil Houlsby, and Lucas Beyer. Scaling vision transformers. In *Proceedings of the IEEE/CVF Conference on Computer Vision and Pattern Recognition*, pp. 12104–12113, 2022.

Chongzhi Zhang, Mingyuan Zhang, Shanghang Zhang, Daisheng Jin, Qiang Zhou, Zhongang Cai, Haiyu Zhao, Xianglong Liu, and Ziwei Liu. Delving deep into the generalization of vision transformers under distribution shifts. In *Proceedings of the IEEE/CVF Conference on Computer Vision and Pattern Recognition*, pp. 7277–7286, 2022.

Sixiao Zheng, Jiachen Lu, Hengshuang Zhao, Xiatian Zhu, Zekun Luo, Yabiao Wang, Yanwei Fu, Jianfeng Feng, Tao Xiang, Philip HS Torr, et al. Rethinking semantic segmentation from a sequence-to-sequence perspective with transformers. In *Proceedings of the IEEE/CVF conference on computer vision and pattern recognition*, pp. 6881–6890, 2021.

Jinghao Zhou, Chen Wei, Huiyu Wang, Wei Shen, Cihang Xie, Alan Yuille, and Tao Kong. ibot: Image bert pre-training with online tokenizer. *arXiv preprint arXiv:2111.07832*, 2021.

Chengxu Zhuang, Alex Lin Zhai, and Daniel Yamins. Local aggregation for unsupervised learning of visual embeddings. In *Proceedings of the IEEE/CVF International Conference on Computer Vision*, pp. 6002–6012, 2019.

