# OpenReview forum: "The Counterattack of CNNs in Self-Supervised Learning: Larger Kernel Size might be All You Need"
_TMLR — Withdrawn by Authors_

### Review · Reviewer_jjn6 · 2023-08-11

**Summary Of Contributions:**

This paper propose to use larger convolution kernel and add an additional batch normalization on ConvNext to shrink the performance gap between ConvNext and Transformer.

**Audience:**

Yes

**Broader Impact Concerns:**

No concern.

**Claims And Evidence:**

Yes

**Requested Changes:**

1. The authors claim larger kernel helps CNN to achieve better performance. But Figure 2 shows further increasing the kernel size to 15x15 decreases the performance. Does that mean larger kernel doesn't necessarily lead to better performance? Any explanation on this?
2. I'm not sure if the method can be generalized to other CNN model architectures besides ConvNext. Although the author mentioned they took ConvNext as the baseline because it's the strongest existing baselines, the improvement brought by the two techniques are not obvious. The author should either run multiple times with different random seeds to show the variance and test other CNN architecture to see if the two techniques can consistently bring improvement.
3. The author admitted their paper doesn't have novelty and is an empirical study. Therefore, either showing their proposed method can have improvement for general CNN architectures or show a significant improvement on certain tasks beyond the margin that could be obtained by simple hyper-parameter tuning would be helpful to convince the audience to try their techniques.

**Strengths And Weaknesses:**

Strengths: The paper shows an interesting perspective that increasing the kernel size and adding additional batch normalization may help increase the performance of the CNNs a bit.
Weaknesses: The proposed method is like a simple hyper-parameter tuning technique. The improvement is not significant and the experiments can't prove these two techniques can be applied to general cases. Actually, in the Figure 2, it already shows that larger kernel of 15x15 decreases the performance than 9x9.

---

### Review · Reviewer_npHs · 2023-08-21

**Summary Of Contributions:**

This paper conduct experiments to show that using attention-free CNN SSL architectures can perform on par with or better than the SSL-trained Transformers. They add two adaptions. The first adaptation is scaling the kernel size up. The second is adding Batchnorm layers after depthwise convolutions. They also find that BC-SSL achieves good results when transferring to downstream tasks. The robustness of BC-SSL improves as the kernel size scales up.

**Audience:**

No

**Claims And Evidence:**

No

**Requested Changes:**

See Weaknesses

**Strengths And Weaknesses:**

Strengths
1.	The paper is generally well written and quite easy to follow.
2.	The empirical results seem good on most of the performed tasks.
3.	The studied problem is interesting.

Weaknesses
1. A significant weakness is the lack of in-depth theoretical analysis of ading BatchNorm after large depthwise kernels and navely scaling up convolutional kernel sizes. Why do these two methods work? Lack of analysis and clear motivation makes the paper more like an experimental report.
2. ading BatchNorm after large depthwise kernels and navely scaling up convolutional kernel sizes are quite straight forward. These more like tricks or hyperparameters tuning.
3. Downstream experiments are only conducted on Mask-RCNN. More evaluated architectures will be better.

---

### Review · Reviewer_u9kc · 2023-08-22

**Summary Of Contributions:**

- Recent advances in SSL have typically used ViTs as the feature extractor which have typically shown better performance compared to CNNs on various downstream tasks.
- The authors claim that this is likely due to use of the vanilla ResNets and the field has since proposed improved variants which have shown much better performance for supervised learning.
- This paper investigates SSL on such "modern" CNNs for SSL and compare them to transformers. The paper shows that these CNNs perform on par or better than transformers for SSL and subsequent evaluation on COCO downstream evaluation.

**Audience:**

Yes

**Claims And Evidence:**

Yes

**Requested Changes:**

Simplicity is a strength of this paper, and I think even the current analysis will be useful to the community. That said, more downstream evaluations and some more downstream tasks would help secure my recommendation.

**Strengths And Weaknesses:**

Strengths:
- The paper is well written and easy to follow.
- The paper is upfront about not proposing any new method, but the presented results are interesting and will be beneficial to the field and practitioners.

Weaknesses:
While the presented SSL pretext task and selection of COCO as a downstream task is interesting, the following would help strengthen the paper and increase its reach :

- DINO is very impressive technique, but some similar experiments on other SSL tasks like Contrastive learning, MAEs would be very interesting.
- While ImageNet performance does show correlation on various downstream tasks, it would still be interesting to evaluate how CNNs fare with transformers on a wider set of downstream evaluations : for example results in [a].

[a] How Well Do Self-Supervised Models Transfer?

---

### Review · Reviewer_SYjJ · 2023-08-23

**Summary Of Contributions:**

This paper explores existing self-supervised learning methods for convolutional neural network backbones, especially in the case of large kernels. Firstly, the author emphasizes that CNN models cannot outperform Vision Transformers (ViT). The author then proposes the use of scaling up the kernel size and adding batch normalization after depthwise layers to improve CNN performance in self-supervised learning.

**Audience:**

Yes

**Claims And Evidence:**

Yes

**Requested Changes:**

Na

**Strengths And Weaknesses:**

Strengths:

This paper is well-written.

The results could benefit the improvement of performance in resource-constrained models.


Weaknesses:

The approach of scaling up the kernel size and adding batch normalization after depthwise layers has been investigated in supervised learning.

This paper doesn't propose novel methods; it solely focuses on the analysis of Self-Supervised Learning (SSL) on CNN models. The recent SSL method, MAE, should be considered by the authors. They could conduct experiments or include a discussion of MAE in the experiments.

---

### Note · Authors · 2023-08-28

I have read and agree with the venue's withdrawal policy on behalf of myself and my co-authors.